# Cannabinoid-Inspired Inhibitors of the SARS-CoV-2 Coronavirus 2′-*O*-Methyltransferase (2′-*O*-MTase) Non-Structural Protein (Nsp10–16)

**DOI:** 10.3390/molecules29215081

**Published:** 2024-10-28

**Authors:** Menny M. Benjamin, George S. Hanna, Cody F. Dickinson, Yeun-Mun Choo, Xiaojuan Wang, Jessica A. Downs-Bowen, Ramyani De, Tamara R. McBrayer, Raymond F. Schinazi, Sarah E. Nielson, Joan M. Hevel, Pankaj Pandey, Robert J. Doerksen, Danyelle M. Townsend, Jie Zhang, Zhiwei Ye, Scott Wyer, Lucas Bialousow, Mark T. Hamann

**Affiliations:** 1Department of Drug Discovery & Biomedical Sciences, Medical University of South Carolina, 280 Calhoun St, Charleston, SC 29425, USA; benjamim@musc.edu (M.M.B.);; 2Department of Chemistry, University of Malaya, Kuala Lumpur 50603, Malaysia; 3Department of Pharmacy, Lanzhou University, Lanzhou 730000, China; 4Center for ViroScience and Cure, Laboratory of Biochemical Pharmacology, Department of Pediatrics, Emory University School of Medicine and Children’s Healthcare of Atlanta, 1760 Haygood Drive, HSRB-1, Atlanta, GA 30322, USA; 5Department of Chemistry & Biochemistry, Logan, UT 84322, USA; 6National Center for Natural Products Research, School of Pharmacy, University of Mississippi, University, MS 38677, USA; 7Department of BioMolecular Sciences and Research Institute of Pharmaceutical Sciences, University of Mississippi, University, MS 38677, USA; 8Department of Cell and Molecular Pharmacology and Experimental Therapeutics, Medical University of South Carolina, 70 President St, DD410, Charleston, SC 29425, USA; 9Department of Public Health Sciences, Medical University of South Carolina,135 Cannon St, Charleston, SC 29425, USA

**Keywords:** cannabinoid, coronavirus, SARS-CoV-2, 2′-*O*-MTase, methyltransferase, Nsp, non-structural protein, in silico, computational screen, natural products, DP4+ NMR analyses

## Abstract

The design and synthesis of antiviral compounds were guided by computationally predicted data against highly conserved non-structural proteins (Nsps) of the SARS-CoV-2 coronavirus. Chromenephenylmethanone-1 (CPM-1), a novel biphenylpyran (BPP), was selected from a unique natural product library based on in silico docking scores to coronavirus Nsps with high specificity to the methyltransferase protein (2′-*O*-MTase, Nsp10–16), which is responsible for viral mRNA maturation and host innate immune response evasion. To target the 2′-*O*-MTase, CPM-1, along with intermediate BPP regioisomers, tetrahydrophenylmethanones (TPMs), were synthesized and structurally validated via nuclear magnetic resonance (NMR) data and DP4+ structure probability analyses. To investigate the activity of these BPPs, the following in vitro assays were conducted: SARS-CoV-2 inhibition, biochemical target validation, mutagenicity, and cytotoxicity. CPM-1 possessed notable activity against SARS-CoV-2 with 98.9% inhibition at 10 µM and an EC_50_ of 7.65 µM, as well as inhibition of SARS-CoV-2’s 2′-*O*-MTase (expressed and purified) with an IC_50_ of 1.5 ± 0.2 µM. In addition, CPM-1 revealed no cytotoxicity (CC_50_ of >100 µM) or mutagenicity (no frameshift or base-pair mutations). This study demonstrates the potential of computational modeling for the discovery of natural product prototypes followed by the design and synthesis of drug leads to inhibit the SARS-CoV-2 2′-*O*-MTase.

## 1. Introduction

The date was 11 March 2020, when the World Health Organization (WHO) declared coronavirus disease 2019 (COVID-19) a pandemic. The mysterious coronavirus that initially spread from Wuhan, China, has since spread worldwide, with nearly 800 million cases and over seven million cumulative deaths, as reported by the WHO—updated October of 2024 [1]. The COVID-19 pandemic sparked a global effort to rapidly develop antiviral therapeutics for those infected by the severe acute respiratory syndrome coronavirus 2 (SARS-CoV-2). As a result, novel messenger RNA (mRNA) vaccines from Pfizer and Moderna, as well as a traditional viral vector vaccine from Johnson and Johnson (J&J), were developed in record-breaking time—largely thanks to years of prior research, modern biotechnology, and Operation Warp Speed [2]. Unlike current vaccines, which immunize the body by inducing the generation of antibodies for future protection, antiviral agents that directly target viral nonstructural proteins (Nsps) are potential alternative treatments for current and future pandemics.

The protein in SARS and MERS coronaviruses responsible for mRNA maturation and host innate immune response evasion is the 2′-*O*-methyltransferase (2′-*O*-MTase; Nsp10–16; PDB ID: 6W4H, 3R24, and 5YNB), which is an Nsp heterodimer complex highly conserved among all *Betacoronaviruses* [3,4,5]. The 2′-*O*-MTases of SARS coronaviruses share primary amino acid sequences up to 95% and 99% identical to Nsp16 and Nsp10, respectively [6]. A recent study suggests that substrate selectivity is broader in SARS-CoV-2 and determined by the Nsp10 cofactor of the Nsp10-16 complex [7]. This highly conserved nature of amino acids and substrate-binding structures suggests that 2′-*O*-MTase is a potential broad-spectrum target amongst coronaviruses. Targeting the 2′-*O*-MTase would prevent the virus’s mRNA from maturing before translation, which is required to mimic eukaryotic mRNA and avoid detection and degradation by cytosolic ribonucleases (RNases) [8,9]. The 2′-*O*-MTase uses *S*-adenosylmethionine (SAM) to methylate Cap-0-RNA (m7GpppA2′-OH-RNA) into Cap-1-RNA (m7GpppAm2′-O-RNA). Small molecule antivirals that inhibit SAM’s natural role as an active methylator of adenosine may prevent coronaviruses from subverting the induction of interferons (IFNs) and translating their viral RNA [4].

Currently, there are only preclinical and clinical candidates aimed at targeting the Nsp10-16 complex. These include sinefungin, a SAM nucleoside analog; 3-Deazaneplanocin A (DZNep), a SAM cycle inhibitor; and other small molecule inhibitors [10,11]. Broad affinity to multiple Nsps is the major reoccurring issue for many of these experimental therapeutics, thus emphasizing the need for a selective inhibitor with reduced chances of off-target effects that may result in cytotoxicity and unintended mechanisms of action (MoAs). As a result, a candidate compound, CPM-1, was sourced from a unique in-house pool of nearly 500 compounds composed of natural product scaffolds, SAM analogs, and repurposed FDA-approved antivirals used to treat SARS-CoV-2 infections (see Appendix A) [12].

Cannabinoid-like synthetics, such as CPM-1 and analogs, may be potential antiviral Nsp inhibitors due to high computational in silico binding affinities to coronavirus Nsps, as well as in vitro activity, as demonstrated in this study. CPM-1 may also be non-cytotoxic and non-mutagenic due to a unique cannabinol (CBN)-like benzochromenepyran (BCP) core motif (Figure 1). Additionally, CPM-1 may also be non-psychoactive since CBN is a partial agonist of the cannabinoid receptor 1 (CB_1_) receptor with low intrinsic activity [13,14,15]. CB_1_, a central nervous system (CNS) neuroreceptor of the endocannabinoid system, is known to bind with the Δ^9^-THC cannabinoid, thus resulting in psychoactivity [16].

It is important to mention that there are no medically reported fatal overdoses from cannabidiol (CBD) or related cannabinoids, which are recreationally used by 14% or nearly 50 million Americans [17,18,19]. Some during the 2020 COVID-19 pandemic claimed that the use of CBD alleviated symptoms [20,21,22]. Interestingly, some studies demonstrate that CBD potently inhibited SARS-CoV-2 replication in lung epithelial cells while reducing the expression of the ACE2 receptor and pro-inflammatory cytokines [21,23,24]. One study indicated that the MoA of cannabinoid ligands is the formation of stable conformations with Nsp binding pockets responsible for the viral RTC [22]. In another, CBD was found to be associated with preventing viral gene expression by upregulating the host’s IFN signaling pathways that target viral RNA, thus preventing translation [21]. Similar studies have shown potency for both Δ^9^-THC (IC_50_ = 10.3 µM) and CBD (IC_50_ = 7.9 µM) against SARS-CoV-2 using Vero cells, which also demonstrated higher molar concentration doses as potentially safe and non-cytotoxic [22,25,26].

In the present study, cannabinoid-inspired synthetic antivirals were designed, screened, and synthesized to target the 2′-*O*-MTase in *Coronaviridae*, specifically SARS-CoV-2. These desired antivirals were chosen based on unique, natural-product-inspired scaffolds, synthetic feasibility, and high computational binding affinity and selectivity to Nsp10–16.

## 2. Results and Discussion

### 2.1. In Silico Screening

Logistical and natural product sourcing challenges resulted in synthetic alternatives to natural product scaffolds for analog production. As a result, CPM-1, a cannabinoid-like synthetic compound, was selected due to computational broad-spectrum coronavirus 2′-*O*-MTase selectivity but with the most promising binding affinity to SARS-CoV-2 of −9.8 kcal/mol (Table 1 and Appendix A). The synthesis of CPM-1 resulted in the generation of four novel molecules of two novel classes: tetrahydrophenylmethanones (TPMs): TPM-1 and TPM-2 and chromenephenylmethanones (CPMs): CPM-1 and CPM-2. These all demonstrated high computational docking scores in comparison to SAM and cannabinoids sharing core motifs with the synthetic cannabinoid-like molecules.

As can be seen in Figure 1, these core motif structures are naturally found in certain phytocannabinoids, such as Δ^8^-THC (delta-8-tetrahyd rocannabinol) and cannabinol (CBN), which are both secondary metabolites found in marijuana (*Cannabis sativa* and *C. indica*) and hemp (*C. sativa*) [27]. *C. indica* is often characterized as a relaxing and pain-relieving species with a more balanced cannabinoid profile, whereas *C. sativa* is often characterized as an energizing and uplifting species with a higher THC content [28].

On the other hand, (−)-*cis*-perrottetinene (PET), a bibenzyl/stilbene cannabinoid from the Radula genus of liverwort plants, shares a tetrahydrodibenzopyran (TBP) scaffold with delta-9-tetrahydrocannabinol (Δ^9^-THC), as seen in Figure 2. This is an example of convergent evolution amongst distinct plant species that share TBP and BCP scaffolds, which were once perceived as exclusive to the *Cannabis* genus [29]. These were all screened as natural product cannabinoids in silico controls.

### 2.2. Synthesis of CPM-1 and Regioisomer Analogs

To produce CPM-1, a synthetic chemical reaction between (*S*)-*cis*-verbenol and a resorcinol moiety was determined as the most straightforward and cost-effective option for TBP motif synthesis. This would be followed by an aromatization reaction to yield a BCP motif [30]. (*S*)-*cis*-verbenol is naturally found in the feverfew flowering plant (*Tanacetum parthenium*) and a few other organisms, whereas many types of resorcinols are synthetics or semi-synthetics produced from plant resins [31,32]. Resorcinols with biphenyl core motifs (e.g., benzophenones and prenyl stilbenes) react with verbenol to produce biphenylpyrans (BPPs). The two-step synthesis of cannabinoid derivatives used in this study to produce a TBP core motif (found in Δ^8^-THC) and a BCP core motif (found in CBN) is outlined in Figure 1.

The first step in the synthesis (Figure 1A) involves a Friedel–Crafts alkylation reaction to form a TBP motif, which is the core ring system of Δ^8^-THC (Figure 1). The second step (Figure 1B) involves an aromatization reaction of the terpenoid cyclohexane from the TBP motif to form a BCP motif, which is the core ring system of CBN. The previously synthesized TPM regioisomers were used to produce CPM regioisomers via the terpenoid aromatization reaction. A total of four BPP products were synthesized: two TPMs (TPM-1 and TPM-2) and two CPMs (CPM-1 and CPM-2). 

### 2.3. DP4+ Probability Analyses

The structures of the BPPs (Figure 1; Appendix A) were confirmed via DP4+ structure probability analyses by comparing experimental and calculated ^1^H and ^13^C NMR spectra data (Appendix A). DP4+ probability analyses were performed to predict the correct regioisomers of the synthesized TPM and CPM compounds [33]. The DP4+ probability analyses revealed that TPM-1 (6a*R*,10a*R*), TPM-2 (6a*R*,10a*R*), CPM-1, and CPM-2 showed excellent agreement (100% probability) with the experimental NMR data of compounds TPM-1, TPM-2, CPM-1, and CPM-2, respectively. In addition, the standard statistical parameters of carbon and proton data, such as mean absolute error (MAE) and corrected MAE, also match the DP4+ probability data with some exceptions. Figure 3 below portrays the DP4+ outcome for CPM-1 compared to the other regioisomers, CPM-2 and CPM-3.

### 2.4. In Vitro Activity

Anti-SARS-CoV-2 activity was assessed by measuring percent inhibition at 10 µM in Vero cells, which ranged from 98.1 to 99.3% inhibition of SARS-CoV-2 *in vitro*, as seen in Table 2. In addition, EC_50_ data were also obtained via antiviral dose–response assays for the four synthesized BPPs, which ranged from 4.3 to 8.6 µM. The four synthetics were nearly as potent as the control, remdesivir, which is a repurposed drug that became the first FDA-approved antiviral medication for COVID-19 [34,35,36].

A cytotoxicity assessment of the four BPP synthetics was carried out on PBM and Vero cells, as seen in Table 3. CPM-1 was the only synthesized BPP with an acceptable, ”safe” cytotoxicity of CC_50_ of >100 µM for both PBM and Vero cells. Interestingly, remdesivir portrayed high cytotoxicity (2.0 µM) in PBM cells. Although Vero cells are non-human, they are regularly utilized for initial toxicity screening. Additional antiviral and cytotoxicity testing is planned via HepG2, a hepatic cell line, and A559, the human lung cell line, for future synthetic cannabinoid-inspired analogs [38,39].

The ability of the four BPPs to inhibit the purified SARS-CoV-2 2′-*O*-MTase was determined using an established commercially available kinetic assay [40,41,42]. The SARS-CoV-2 2′-*O*-MTase is composed of two proteins, Nsp10 and Nsp16, both of which were expressed and purified according to previous work [9,42]. Enzyme activity was measured under linear conditions with Cap-0 (m7GpppAUUAA) RNA as a substrate, as described in the Appendix A. All the synthesized compounds demonstrated inhibition of 2′-*O*-MTase, ranging from 1.5 to 6.7 µM (Table 4 and Appendix A). For comparison, a control nucleoside-based methyltransferase inhibitor, sinefungin, has an IC_50_ of 3.4 ± 0.4 µM [43]. These data reveal that the synthesized molecules, TPM-1 and CPM-1, comparably inhibit the SARS-CoV-2 2′-*O*-MTase as well as or better than the nucleoside-based sinefungin.

Synthesized compound CPM-1 was selected to be assayed for mutagenicity due to its minimal cytotoxicity in vitro and potent inhibition of SARS-CoV-2 2′-*O*-MTase. As a result, the Ames fluctuation test for CPM-1 at different concentrations was utilized to induce a mutagenic index (MI) in two different strains of *Salmonella*. A compound is considered mutagenic if the peak of the average number of reverse mutant colonies of strains TA98 and TA100 is greater than twice the average number of the negative control (DMSO). None of the six concentrations of CPM-1 tested with *Salmonella* TA98 (Figure 4A) or TA100 (Figure 4B) induced a reverse mutation.

### 2.5. Possible Inhibition Mechanisms

A previous study identified the catalytic KDKE tetrad motif (Lys6844-Asp6928-Lys6968-Glu7001) in SARS-CoV-2 2′-*O*-MTase as essential for methyltransferase activity. Asp6928 initiates cap formation and facilitates methyl transfer from SAM to RNA, while the lysine residues stabilize the RNA, and glutamic acid ensures structural integrity. Analysis of the 2′-*O*-MTase-SAM binding domain revealed that Asp6928 forms a hydrogen bond with the N-atom of SAM’s methionine unit [42,44]. For TPMs and CPMs, the binding interactions involve distinct amino acid residues (Appendix A) from that of SAM. However, their location within the narrow binding domain of SAM, in close proximity to the Asp6928 residue, suggests that CPMs and TPMs may inhibit methyltransferase activity by obstructing access to the catalytic KDKE tetrad (Figure 5). This finding aligns with our earlier study on Machaeriols RS-1 and RS-2 [42]. The highly conserved nature of the SAM-dependent 2′-*O*-MTase protein across *Betacoronaviruses*, including SARS-CoV-2, SARS, and MERS, is well established [45]. Our previous study demonstrated that superposition and sequence alignment of the 2′-*O*-MTase protein structures of SARS-CoV-2 (PDB ID: 6W4H), SARS (PDB ID: 3R24), and MERS (PDB ID: 5YNB) revealed similar SAM-binding domains, all of which share the catalytic KDKE tetrad [42]. Thus, TPMs and CPMs may serve as potential broad-spectrum inhibitors of 2′-*O*-MTase in *Betacoronaviruses*.

## 3. Materials and Methods

### 3.1. Chemicals and Reagents

ACS and HPLC grade chemicals, reagents, and solvents were purchased through VWR (Leicestershire, UK), ThermoFisher Scientific (Waltham, MA, USA), and Sigma Aldrich (St. Louis, MO, USA). Starting reagents for producing TPMs were (*S*)-*cis*-verbenol (95%) and 2,2′,4,4′-tetrahydroxybenzophenone (97%), along with tetrafluoroboric acid-diethyl ether (HBF_4_·OEt_2_), used as a reaction catalyst. (*S*)-*cis*-verbenol is naturally found in the feverfew flowering plant (*Tanacetum parthenium*) and a few other organisms, whereas many types of resorcinols, such as TBP, are synthetics produced from plant resins [31,32]. (*S*)-*cis*-verbenol has the “*S*” configuration and has the specific rotation ${\left[\alpha \right]}_{D}^{20}$ = −9° in chloroform. A vanillin stain solution (15 g/250 mL EtOH + 2.5 mL H_2_SO_4_), in conjunction with a heat gun, was utilized to observe and monitor the reaction and final products. Iodine (I_2_, >99%) was utilized as a reagent for the aromatization reaction. A Fast Blue B Salt solution (0.1 g/250 mL DI H_2_O) was utilized to stain CPM compounds possessing aromatic rings. Reactions were monitored for completion via TLC.

Plasmids encoding the Nsp10 and Nsp16 subunits were obtained from BEI Resources (NR-52425 and NR-52427). Cap-0 (m7GpppAUUAA) mRNA was synthesized by Bio-Synthesis, Inc. (Lewisville, TX, USA).

### 3.2. Chemistry

All solvents and chemicals were used as purchased without further purification. All moisture and air-sensitive reactions were performed in an inert atmosphere via argon gas in oven-dried or flame-dried glassware. For this experiment, dried Schlenk tubes were utilized as a reaction vessel. Reactions that required heating were carried out with a stir-hot plate using a heated external oil bath. The progress of all reactions was monitored on Merck precoated silica gel plates (with fluorescence indicator UV_254_) using ethyl acetate/n-hexane as a solvent system. Column chromatography was performed with SiliaFlash silica gel P60 (230–400 mesh), with the solvent mixtures specified in the corresponding experiment. Spots were visualized by irradiation with ultraviolet light (254 nm). The synthesized compounds were isolated by high-performance liquid chromatography (HPLC). The instrument was a Waters^TM^ 486 Tunable Absorbance Detector and Automated Gradient Controller (Waters Corp., Milford, MA, USA), and the column was a Kinetex^®^ LC C18 100 Å, 250 × 21.2 mm column [Phenomenex, Torrance, CA, USA]. A program gradient, with a run time of 45 min and a flow rate of 7 mL/min, consisted of an initial gradient of 75% H_2_O (DI H_2_O + 0.1% formic acid) and 25% MeOH (HPLC-grade), flowing to 0% H_2_O and 100% MeOH. All compounds were determined to be ≥95% pure by HPLC analysis. Proton (^1^H) and carbon (^13^C) NMR spectra were recorded on a Bruker Avance^TM^ II 600 MHz with UltraShield^TM^ Plus magnet technology (Bruker BioSpin, Rheinstetten, Germany), and a probe temperature of 307 K. Samples were dissolved using deuterated chloroform (CDCl_3_) as a solvent. Chemical shifts are given in parts per million (ppm) (δ relative to residual solvent peak for ^1^H and ^13^C). For ^1^H NMR spectra, the proton signal (ppm) was at 7.28. For ^13^C NMR spectra, the carbon signals were at 76.8, 77.0, and 77.20. One-dimensional (1D) ^1^H NMR data was collected via the “PROTON” parameter and “zg30” pulse sequence, and ^13^C NMR data was collected via “C13CPD” parameter and “zgpg30” pulse sequence. Mass spectrometry (MS) data were collected via Impact II Elute QTOF UPLC (Bruker Daltonics, Bremen, Germany). Data were analyzed using DataAnalysis^®^.

#### 3.2.1. Synthesis of TPM-1 and TPM-2

To a stirred solution of 2,2′,4,4′-tetrahydrobenzophenone resorcinol (246 mg, 1.0 mmol, 1.0 equiv.) in dry acetone (2 mL) at −78 °C was added HBF_4_∙OEt_2_ (0.3 mL) dropwise. A solution of (*S*)-*cis*-verbenol (183 mg, 1.2 mmol, 1.2 equiv.) in DCM (4 mL) was added dropwise to the reaction mixture and kept at −78 °C for 2 hrs. After removing from the cold bath and at room temperature for 1 hr., the reaction was quenched by the addition of sat. aq. NaHCO_3_. The phases were separated, and the organics were washed with NaHCO_3_ (×3). The aqueous phase was extracted with dry DCM (×3). The combined organic extracts were dried over anhydrous Na_2_SO_4_. After filtration, the solvent was removed in vacuo, and the products were isolated from the crude reaction mixture by normal phase silica gel column chromatography (gradient: 5%, 20%, 50%, and 100% ethyl acetate in hexanes as eluent) as yellow oils. The 20% ethyl acetate fraction was then further fractionated via an isocratic elution using silica gel. Further purification was achieved by reverse phase HPLC (gradient: 75% H_2_O/25% MeOH to 0% H_2_O/100% MeOH). The final yields of TPM-1 and TPM-2 were 15.2% (65.0 mg) and 10.6% (45.5 mg), respectively. Prior to HPLC, the fraction was injected into a small disposable cartridge: a C18-E SPE sorbent (attached to a 13 mm syringe filter with a 0.2 μM PTFE membrane).

TPM-1 (2,4-dihydroxyphenyl)((6a*R*,10a*R*)-1-hydroxy-6,6,9-trimethyl-6a,7,10,10a-tetrahydro-6*H*-benzo[c]chromen-2-yl)methanone: off-white solid; yield 15.2%; ^1^H NMR (600 MHz, CDCl_3_ 7.28 ppm): δ = 12.04 (s, 1H, OH), 11.13 (s, 1H, OH), 7.53 (d, *J* = 8.7 Hz, 1H, Ar-H), 7.39 (d, *J* = 8.9 Hz, 1H, Ar-H), 6.47 (d, *J* = 2.5 Hz, 1H, Ar-H), 6.40 (dd, *J* = 8.7, 2.5 Hz, 1H, Ar-H), 6.37 (d, *J* = 8.9 Hz, 1H, Ar-H), 5.54 (brs, 1H, OH), 5.46–5.43 (m, 1H, CH=), 3.36 (dd, *J* = 17.1, 4.1 Hz, 1H, CH_2_), 2.80 (td, *J* = 10.9, 4.8 Hz, 1H, CH), 2.20–2.14 (m, 1H, CH2), 1.91–1.77 (m, 3H, CH_2_), 1.72 (s, 3H, CH_3_), 1.43 (s, 3H, CH_3_), 1.15 ppm (s, 3H, CH_3_). ^13^C NMR (150 MHz, CDCl_3_): δ = 199.5, 164.1, 163.9, 161.6, 160.6, 135.2, 134.9, 132.3, 118.9, 114.2, 114.1, 112.6, 109.4, 107.2, 104.0, 78.6, 44.6, 35.5, 31.5, 27.8, 27.4, 23.4, 18.8 ppm. HPLC analysis: TPM-1 was collected at 4% H_2_O, 96% MeOH on the 43rd minute of the gradient run. MS: 381.208 *m*/*z* [M + H]^+^ for C_23_H_24_O_5_.

TPM-2 (2,4-dihydroxyphenyl)((6aR,10aR)-3-hydroxy-6,6,9-trimethyl-6a,7,10,10a-tetrahydro-6*H*-benzo[c]chromen-2-yl)methanone: pale yellow solid; yield 10.6%; ^1^H NMR (600 MHz, CDCl_3_ 7.28 ppm): δ = 11.30 (s, 1H, OH), 10.88 (s, 1H, OH), 7.52 (d, *J* = 8.7 Hz, 1H, Ar-H), 7.45 (d, *J* = 1.3 Hz, 1H, Ar-H), 6.49 (d, *J* = 2.5 Hz, 1H, Ar-H), 6.43 (dd *J* = 8.7, 2.5 Hz, 1H, Ar-H), 5.53 (brs, 1H, OH), 5.47–5.45 (m, 1H, CH=), 2.69 (td, *J* = 11.4, 5.5 Hz, 1H, CH), 2.49–2.46 (dd, 1H, CH), 2.47 (m, 1H, CH2), 2.20–2.14 (m, 1H, CH), 1.94–1.88 (m, 1H, CH2), 1.87–1.80 (m, 2H, CH2), 1.71 (s, 3H, CH_3_), 1.42 (s, 3H, CH_3_), 1.21 ppm (s, 3H, CH_3_). ^13^C NMR (150 MHz, CDCl_3_): δ = 199.3, 164.5, 162.1, 161.7, 160.2, 135.0, 132.9, 132.2, 119.9, 117.8, 114.2, 113.7, 107.2, 105.2, 104.1, 78.9, 42.6, 36.6, 31.5, 27.5, 27.3, 23.4, 19.7 ppm. HPLC analysis: TPM-2 was collected at 7% H_2_O and 93% MeOH on the 41st minute of the gradient run. MS: 381.212 *m*/*z* [M + H]^+^ for C_23_H_24_O_5_.

#### 3.2.2. Synthesis of CPM-1

To a stirred solution of TPM-1 (52.5 mg, 0.138 mmol) in toluene (5.25 mL, 1 mL/mmol) at 90 °C was added iodine (I_2_) (52.5 mg, 1.0 equiv., 0.207 mmol). After 14.5 h, additional I_2_ (52.5 mg, 1.0. equiv.) was added. After 18 hrs., the reaction mixture was allowed to cool to room temperature and was quenched by the addition of sat. aq. NaHCO_3_. The phases were separated, and the organics were washed with NaHCO_3_ (×3), Na_2_S_2_O_3_ (×3), and brine sat. aq. solutions. The aqueous phase was extracted with hexane (×3), and the combined organic extracts were dried over anhydrous Na_2_SO_4_. After filtration, the solvent was removed in vacuo. Purification was achieved by reverse phase HPLC (gradient: 75% H_2_O/25% MeOH to 0% H_2_O/100% MeOH). The final yield of CPM-1 was 11.8% (6.2 mg). Prior to HPLC, the fraction was injected into a small disposable cartridge: a C18-E SPE sorbent (attached to a 13 mm syringe filter with a 0.2 µM PTFE membrane.

CPM-1 (2,4-dihydroxyphenyl)(1-hydroxy-6,6,9-trimethyl-6*H*-benzo[c]chromen-2-yl)methanone): yellow solid; yield 11.8%; ^1^H NMR (600 MHz, CDCl_3_ 7.28 ppm): δ = 12.45 (s, 1H, OH), 11.13 (s, 1H, OH), 8.47 (s, 1H, Ar-H), 7.55 (d, *J* = 8.7 Hz, 1H, Ar-H), 7.49 (d, *J* = 8.7 Hz, 1H, Ar-H), 7.16 (d, *J* = 7.9 Hz, 1H, Ar-H), 7.13 (dd, *J* = 7.9, 1.6 Hz, 1H, Ar-H), 6.54 (d, *J* = 8.7 Hz, 1H, Ar-H), 6.49 (d, *J* = 2.5 Hz, 1H, Ar-H), 6.43 (dd, *J* = 8.7, 2.5 Hz, 1H, Ar-H), 2.41 (s, 3H, CH_3_), 1.65 ppm (s, 3H, CH_3_), 1.65 (s, 3H, CH_3_) ppm. ^13^C NMR (150 MHz, CDCl_3_): δ = 199.95, 164.4, 162.4, 161.7, 160.2, 137.2, 136.0, 135.3, 133.9, 128.5, 127.5, 126.4, 122.4, 122.4, 114.1, 113.9, 109.5, 107.5, 104.0, 78.9, 27.5, 27.5, 21.6 ppm. HPLC analysis: CPM-1 was collected at 6% H_2_O and 96% MeOH on the 42nd minute of the gradient run. MS: 377.269 *m*/*z* [M + H]^+^ for C_23_H_20_O_5_.

#### 3.2.3. Synthesis of CPM-2

To a stirred solution under reflux conditions of TPM-2 (20 mg, 0.0526 mmol) in toluene (2.7 mL, 1 mL/mmol) at 80 °C was added iodine (I_2_) (20 mg, 1.0 equiv., 0.0788 mmol). After 1.5 hrs., additional I_2_ (20 mg, 1.0. equiv.) was added. The reaction was monitored for completion by TLC (254 nm UV and Fast Blue staining). After 3.5 hrs., the reaction mixture was allowed to cool to room temperature and was quenched by the addition of sat. aq. NaHCO_3_. The phases were separated, and the organic was washed with NaHCO_3_ (×3), Na_2_S_2_O_3_ (×3), and NaCl salt brine (×3) sat. aq. solutions. The aqueous layer was separated from the organic layer during the washing steps and further extracted with hexane. Further purification could be achieved by reverse phase HPLC (gradient: 75% H_2_O/25% MeOH to 0% H_2_O/100% MeOH). The final yield of CPM-2 was 19.0% (3.8 mg). Prior to HPLC, the fraction was injected into a small disposable cartridge: a C18-E SPE sorbent (attached to a 13 mm syringe filter with a 0.2 µM PTFE membrane).

CPM-2 (2,4-dihydroxyphenyl)(3-hydroxy-6,6,9-trimethyl-6*H*-benzo[c]chromen-2-yl)methanone: yellow solid; yield 19.0%; ^1^H NMR (600 MHz, CDCl_3_ 7.28 ppm): δ = 11.31 (s, 1H, OH), 11.06 (s, 1H, OH), 7.96 (d, *J* = 1.5 Hz, 1H, Ar-H), 7.63 (d, *J* = 8.7 Hz, 1H, Ar-H), 7.31 (s, 1H, Ar-H), 7.14 (d, *J* = 8.7 Hz, 1H, Ar-H), 7.10 (dd, *J* = 8.7, 1.5 Hz, 1H, Ar-H), 6.60 (s, 1H, Ar-H), 6.53 (d, *J* = 2.6 Hz, 1H, Ar-H), 6.47 (dd, *J* = 8.7, 2.6 Hz, 1H, Ar-H), 5.54 (brs, 1H, OH), 2.37 (s, 3H, CH3), 1.67 (s, 3H, CH3), 1.67 (s, 3H, CH3) ppm. ^13^C NMR (150 MHz, CDCl3): δ = 199.4, 164.8, 163.9, 162.0, 159.8, 137.7, 135.5, 135.1, 128.5, 127.6, 127.2, 123.4, 121.9, 114.6, 114.1, 107.5, 106.3, 104.2, 79.3, 28.2, 28.2, 21.3 ppm. HPLC analysis: CPM-2 was collected at 10% H_2_O and 90% MeOH on the 41st minute of the gradient run. MS: 377.273 *m*/*z* [M + H]^+^ for C_23_H_24_O_5_.

### 3.3. In Silico Molecular Binding Assay

The compound structures were optimized using MM2 in Chem3D Ultra version 16.0. The crystal structures of the 2′-*O*-MTase (PDB ID: 6W4H (x-center = 83.181, y-center = 16.183, z-center = 28.120), 3R24 (x-center = 57.272, y-center = 62.272, z-center = 68.032), and 5YNB (x-center = 61.628, y-center = 87.066, z-center = 148.084)) were obtained from Protein Data Bank [46]. AutoDockTools version 1.5.6 was used to prepare the receptor proteins and ligands for the molecular docking experiment. The grid box parameters were grid box spacing = 1.0 Å; x-dimension = y-dimension = z-dimension = 20 Å [47,48]. The AutoDock Vina program was used to perform the docking and calculate the binding affinity. Lastly, the results were processed and analyzed using the BIOVIA Discovery Studio Visualizer v21.1.0.20298.

### 3.4. IC_50_ Determination Using MTaseGlo Coupling Assay

RNA methyltransferase activity of the Nsp10–16 complex was measured using the commercially available coupled assay MTaseGlo (Promega Corp., Madison, WI, USA). The MTaseGlo kit measures the byproduct of methyltransferase reactions, *S*-adenosyl homocysteine (SAH), using luminescence. Nsp10–16 methyltransferase activity was assessed with 25 nM heterodimer (Nsp10–16), 1 µM excess Nsp 10, 20 mM Tris-HCl pH 8.0, 1 mM EDTA, 2 mM MgCl_2_, 300 nM Cap-0 (m7GpppAUUAA) RNA, and 5 µM *S*-adenosylmethionine and varying amounts of inhibitor solubilized in DMSO. The control reaction (no inhibitor) also contained 1% DMSO. Reactions were initiated with 2′-*O*-MTase and terminated at 15 min, as described in the manufacturer’s instructions. Previous studies showed that under these conditions, the rate of methyltransferase activity was linear for at least 20 min. The amount of SAH in each reaction was determined using an SAH standard curve and the same coupled assay. Measurements were corrected by subtracting luminescence associated with a response that lacked RNA substrate. Reactions were carried out in at least duplicate. In order to avoid false positives from compounds that interfere with the MTaseGlo coupling system, each inhibitor was evaluated for its ability to alter the SAH standard curve. At 10 µM, none of the inhibitors affected the standard curve. The percent activity at all inhibitor concentrations was determined by taking the rate of Nsp10–16 in the presence of the inhibitor divided by the rate of Nsp10–16 in the absence of the inhibitor and multiplied by 100. Measurements were taken in at least duplicate, and the average for each measurement was plotted as a function of inhibitor concentration. Data were fitted to the Hill equation using guidelines outlined previously [49,50,51].

### 3.5. In Vitro Cytotoxicity Assay

An MTS (3-(4,5-dimethylthiazol-2-yl)-5-(3-carboxymethoxyphenyl)-2-(4-sulfophenyl)-2*H*-tetrazolium) assay was performed on PBM cells and Vero cells using the CellTiter 96^®^ Non-Radioactive Cell Proliferation (Promega) kit as previously described. Briefly, cell proliferation was measured with or without test compounds after two to four days of incubation. Cytotoxicity was expressed as the concentration of test compounds that inhibited cell proliferation by 50% (CC_50_) and calculated using the Chou and Talalay method [52]. The protocol for this assay followed the methods utilized by Zandi et al. in “Repurposing nucleoside analogs for human coronaviruses” [37].

### 3.6. In Vitro Antiviral Assay

An antiviral evaluation assay was conducted to determine the potential antiviral effects of the natural products and synthetics against in vitro replication of SARS-CoV-2 in cell culture. A confluent monolayer of Vero cells in a 96-well cell culture microplate was treated with 10 µM of compounds, followed by inoculation with 0.1 MOI of the virus. To assess the antiviral activity, a virus yield reduction assay using specific qRT-PCR for each virus was performed. A dose-dependent antiviral assay was conducted for BPPs: TPM-1, CPM-1, TPM-2, and CPM-2. Antiviral activity was further confirmed by virus yield reduction assay using specific qRT-PCR for SARS-CoV-2 by measuring the RNA copy number of the virus after 2 days post-treatment for Vero cells in the supernatant of treated-infected cells in a dose-response manner. One-step qRT-PCR was carried out in a final volume of 10 µL containing extracted viral RNA, specific probe/primer mix, and qScript-Tough master mix (Quantibio, Beverly, MA, USA). Quantitative PCR measurement was performed using a LightCycler^®^ 480 PCR system (Roche, Mannheim, Germany) according to the manufacturer’s protocol. The protocol for this assay followed the methods utilized by Zandi et al. [37].

### 3.7. In Vitro Ames Assay

The Ames-MOD ISO™ test protocol was performed according to the method’s supplier protocol (Environmental Bio-Detection Products Inc.). Sodium azide (NaN_3_, 5 µg/mL) and 2-nitrofluorene (2-NF, 300 µg/mL) were used as positive controls for TA100 and TA98, respectively, while dimethyl sulfoxide (DMSO) was used as the negative control. The overnight cultures were diluted to OD600 = 0.05 for TA100 and =0.1 for TA98 with exposure medium. Then, 200 µL of diluted bacterial culture and 200 µL of exposure medium were added to each well. After 100 min of incubation at 37 °C, 1.6 mL bacterial culture from 24-well plates was mixed with 8.71 mL reversion medium and then transferred into twelve 96-well plates for each strain evaluated. Plates were sealed into Ziploc bags and incubated for 4 days at 37 °C. The plates were scored visually: yellow and partial yellow wells were scored as positive; purple wells were scored as negative.

### 3.8. DP4+ Probability Statistical Method and Calculations

The 2D structures of all plausible regioisomers of TPM and CPM compounds (TPM-1 (6a*R*,10a*R*), TPM-2 (6a*R*,10a*R*), TPM-3 (6a*R*,10a*R*), TPM-1 (6a*R*,10a*S*), TPM-2 (6a*R*,10a*S*), TPM-3 (6a*R*,10a*S*), CPM-1, CPM-2, and CPM-3) were sketched in Maestro (Maestro, Schrödinger, LLC, New York, NY, 2020) and 3D-energy minimized at physiological pH 7.4 using the LigPrep (Schrödinger software Release 2020-4 LigPrep, Schrödinger, LLC, New York, NY, USA) module of the Schrödinger software. The conformational searches of each regioisomer were performed using MacroModel, considering mixed torsional/low-mode sampling. The energy window cutoff was set to 10 kcal/mol to cover all possible lowest energy conformers. Redundant conformers were eliminated using RMSD cutoff = 0.5 Å. The conformations that showed >1% Boltzmann population from molecular mechanics calculations were further geometry optimized using DFT with *m*PW1PW91/6-311+G(d,p), using Gaussian 16 Rev. B.01 software (Gaussian, Inc., Wallingford, CT, USA, 2016). CHCl_3_ was used as a solvent with the PCM solvation model [53]. All the geometry optimizations included subsequent frequency calculations to verify that true minima on the potential energy surface were obtained. The Boltzmann-weighted optimized low-energy conformers in CHCl_3_ were used in the chemical shift calculations using GIAO NMR at the DFT *m*PW1PW91/6-311+G(d,p) level [54]. 

### 3.9. Statistics

The half-maximal effective concentration (EC_50_) for antiviral activity was calculated using GraphPad PRISM for Windows, version 9 (GraphPad Software Inc., San Diego, CA, USA, 2005). In addition, for the Ames test, a sample is considered mutagenic when there is a significant increase in the number of positive wells in treated plates over those found in the negative control plates (i.e., mutagenic index (MI)). The results were expressed as Mutagenicity Ratio (MR = number of positive wells in treated plates/number of positive wells in control plates).

## 4. Conclusions

Currently, there are no FDA-approved COVID-19 therapeutics that target the coronavirus 2′-*O*-MTase. Existing protease and RdRp-inhibiting antivirals fail to prevent infection in the first place, as they are designed to mitigate severe symptoms following infection and, in many cases, after one has been hospitalized. They are unable to mechanistically: (1) prevent initially replicated viral RNA from being processed into infectious virions; (2) stimulate a host innate immune response to attack foreign RNA in human cells; or (3) reduce the risk of viral RNA mutations, resulting in SARS-CoV-2 variants. This emphasizes the need for antivirals that prevent viral replication in the first place, such as a drug that can be administered following exposure or upon anticipation of infection. A 2′-*O*-MTase inhibitor would theoretically prevent a coronavirus from initiating the viral cycle and can be utilized in combination with an alternative Nsp inhibitor (e.g., RdRp) or a vaccine, which will ultimately reduce the virulence of a virus and hinder antiviral resistance mechanisms.

As a result, two sets of novel BPP regioisomeric classes, TPMs and CPMs, were selected for chemical synthesis due to compelling in silico evidence demonstrating broad-spectrum binding affinity to the coronavirus 2′-*O*-MTase and the benign track record of recreational cannabinoid use. Recent evidence from the literature suggests that Nsp16 plays an essential role in driving coronavirus RNA capping, especially in SARS-CoV-2, which may be the MoA of the synthesized set of molecules [55]. TPM-1, CPM-1, TPM-2, and CPM-2 share core TBP or BCP scaffolds present in certain cannabinoids (e.g., Δ^8^-THC and CBN) found in nature that may conform with high binding affinity as antagonists to the 2′-*O*-MTase binding pocket. In vitro evaluation confirmed the inhibitory activity of these cannabinoid-inspired synthetics against the Nsp10–16 complex and the SARS-CoV-2 virus.

The non-cytotoxic and non-mutagenic properties of CPM-1, in particular, demonstrate the potential for a candidate antiviral against 2′-*O*-MTase mutants, which would be the first therapeutic of its kind. CPM-1 possesses a BCP cannabinoid-like motif that is likely to share many properties with CBD and CBD-rich extracts that are non-mutagenic and non-genotoxic yet have displayed cytotoxicity in HepG2 cells [56]. CPM-1 possesses a unique aromatized BCP ring system that may be non-cytotoxic to human cells, unlike other cannabinoids, and may also be non-psychoactive due to the low likelihood of agonizing the CB_1_. Although the natural-product-inspired molecules presented in this research were designed to target the 2′-*O*-MTase, synthetic cannabinoid receptor agonists (SCRAs) designed to target CB_1_ exist, and their pharmacology and toxicology are poorly understood. As a result, future cytotoxicity and pharmacology studies, such as primary binding and functional assays, are planned for the cannabinoid class via the National Institute of Mental Health Psychoactive Drug Screening Program (NIMH PDSP). In addition, CPM-1 has garnered attention from the National Institute of Allergy and Infectious Diseases (NIAID) for in vivo studies. Future directions entail further toxicity and pharmacokinetic studies, as well as the production of additional analogs that follow the modern and streamlined approach presented in this manuscript for candidate drug selection.

The design and synthesis of CPM-1 and its regioisomers demonstrate the potential of utilizing natural products as scaffolds and analogs for in silico screening of drug targets. Nature has undoubtedly been the greatest source for structurally diverse lead drugs/therapeutics for infectious diseases [57]. Furthermore, rapid advancements in AI-driven screening and optimization will generate vast drug leads and will help solve MoAs and structure–activity relationships (SARs) [58].

## Data Availability

The data presented in this study are available in the article.

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
