# Peer review of "Cannabinoid-Inspired Inhibitors of the SARS-CoV-2 Coronavirus 2′-O-Methyltransferase (2′-O-MTase) Non-Structural Protein (Nsp10–16)"

_molecules, 2024, doi:10.3390/molecules29215081_

Round 1

Reviewer 1 Report

Comments and Suggestions for Authors

Manuscript by Menny M. Benjamin et. al. entitled  Cannabinoid-Inspired Inhibitors of the SARS-CoV-2 Corona-2 virus 2-O-Methyltransferase (2′-O-MTase) Nonstructural Protein (Nsp10–16)describes synthesis of  only four cannabiod-like compounds (TPM-1,2 and CPM-1,2) that were selected based on in silico binding affinities for SARS-CoV-2 Nsps, in vitro anti-SARS-CoV-2 of synthesized compounds in Vero cells, inhibition of SARS-CoV-2 2′-O-MTase and cytotoxicity assessment on PBM and Vero cells.

 In the Introduction, it is necessary to add from the last part of the chapter Results and discussion (2.3. In vitro activity (lines 189-205)) since it is more suitable for the Introduction.

In the Results and Discussion the authors self-cite references (eg. 14-19) that is not relevant and fully applicable in context of this research. Also, lines 189-205 should be omitted from this chapter and included in the Introduction. It is necessary to remove scheme 1 because it is completely identical to that shown in scheme 2.

The main remark refers to the confirmation of the structure of the synthesized compounds with which in vitro tests were performed.

The synthesized compounds were structurally characterized by NMR spectroscopy, but in the listed chemical shifts in 1H and 13C NMR spectra signal intensities are misunderstandings.  

It is recommended to perform an analysis of the NOESY spectra, in order to irrefutably determine the structure of the compounds. Thus, from the NOESY spectrum, the interaction between the proton OH and 9a would be evident, for example in the compound TPM-1, while this would be absent in the compound TPM-2.

Many technical corrections and suggestions related to structure confirmation are provided in the PDF manuscript.

In this form the manuscript is not acceptable for publication in a journal such as Molecules. Therefore, I suggest minor revision of the manuscript before publishing. 

Author Response

Please see the attached file. In addition, please see the supplemental documents uploaded to the revised manuscript section.

Reviewer 2 Report

Comments and Suggestions for Authors

This article elaborates on the research process and results of designing and synthesizing new biphenylpyran inhibitors targeting the SARS-CoV-2 coronavirus 2′-O-MTase based on computational prediction data. The author's idea of using computational modeling data to guide the design and synthesis of antiviral drugs and conducting research on the highly conserved non-structural proteins of the SARS-CoV - 2 coronavirus is innovative. 1. In the results section, the content of Table 3 should not be between Table 2 and Table 4. The content of Table 3 should independently discuss the inhibitory results of compounds on the methyltransferase protein (2 - O - MTase, Nsp10 - 16). 2. The article does not provide characterizations or other data such as immunoblotting and RNA nucleic acid electrophoresis to prove the inhibitory results of compounds on methyltransferase. Only the IC50 values are listed, indicating the inhibitory effect. 3. In the ending part of the results, the author believes that the non-cytotoxic and non-mutagenic properties of CPM - 1 are related to cannabis abuse. This view needs to be modified because the source of Vero (African green monkey kidney cells) is not human. 4. The non - cytotoxic and non - mutagenic properties of CPM - 1 need to be further discussed. 5. The HRMS or elemental analysis for the synthesized compounds should be added in the experimental. 6. This article should supplement the prospects for clinical applications, including whether there are plans for in vivo experimental studies in the future, as well as the preliminary research ideas and directions. It should also analyze the potential advantages and challenges that these compounds may face in clinical applications, providing references for subsequent research.

Comments on the Quality of English Language

Moderate editing of English language required.

Author Response

Please see the attached file. In addition, please see the SI documents attached to the revised manuscript section.

Reviewer 3 Report

Comments and Suggestions for Authors

The article submitted for review is about the search for new substances with activity against Sars-Cov-2 coronavirus. It shows a very modern approach - starting from plant substances using computer methods, through synthesis and testing of biological activity. The article shows such a comprehensive approach described in an almost exemplary manner.

The introduction describes the problem of the coronavirus pandemic in a comprehensive and precise manner justifies the need to search for substances with antiviral activity.

Can the authors justify the selection of cannabinoids as the backbone in the search for compounds? Why were these compounds chosen? Were derivatives of other cannabinoids also considered?

Please note the differences of the two species - Cannabis sativa and Cannabis indica. Cannabis sativa is referred to as hemp and indica as marijuana.

Why did the authors limit themselves to only stating the binding energy from docking? I suggest enriching the article with an analysis of the binding site, conformation and interactions at the active site. Energy values alone are not very informative about the interaction with the protein.

I cannot analyse the 1H and 13C NMR data from the supplementary data - I do not see the included supplementary materials in the system.

Bioassays - Anti-SARS-CoV-2 Activity, cytotoxicity - no statistics. Have these calculations been performed?
Have statistical differences between Anova test results been assessed?

Please note the editorial aspects - indexes in the 1H NMR assay, and the method of citation required in the journal - dot after the reference number e.g. [2]. instead of .[2]

Author Response

(The authors gave the same response as above.)

Round 2

Reviewer 2 Report

Comments and Suggestions for Authors

This article elaborates on the research process and results of designing and synthesizing new biphenylpyran inhibitors targeting the SARS-CoV-2 coronavirus 2′-O-MTase based on computational prediction data. Most of the comments have been addressed and revised, and so the publication is recommended.

Comments on the Quality of English Language

No

Reviewer 3 Report

Comments and Suggestions for Authors

The Authors improved the quality of their manuscript. It can be accepted in its present form in my opinion.